# *Alr* Gene in *Brucella suis* S2: Its Role in Lipopolysaccharide Biosynthesis and Bacterial Virulence in RAW264.7

**DOI:** 10.3390/ijms241310744

**Published:** 2023-06-28

**Authors:** Mingyue Hao, Minghui Wang, Danyu Zhao, Yong Shi, Ye Yuan, Junmei Li, Yunyi Zhai, Xiaofang Liu, Dong Zhou, Huatao Chen, Pengfei Lin, Keqiong Tang, Wei Liu, Yaping Jin, Aihua Wang

**Affiliations:** 1College of Veterinary Medicine, Northwest A&F University, Yangling District, Xianyang 712100, China; haoshuaifirst@nwafu.edu.cn (M.H.); wangmh@nwafu.edu.cn (M.W.); zhaodanyu@nwafu.edu.cn (D.Z.); hongxuelange@126.com (Y.S.); 15637737955@163.com (Y.Y.); junmei_li@nwafu.edu.cn (J.L.); zhaiyunyi@nwafu.edu.cn (Y.Z.); liuxf@nwafu.edu.cn (X.L.); zhoudong1949@163.com (D.Z.); htchen@nwafu.edu.cn (H.C.); linpengfei@nwsuaf.edu.cn (P.L.); tangkeqiong20036@163.com (K.T.); wliu20cn@yahoo.com (W.L.); 2Key Laboratory of Animal Biotechnology of the Ministry of Agriculture, Northwest A&F University, Yangling District, Xianyang 712100, China

**Keywords:** *Brucella suis* S2, alanine racemase, *alr*, lipopolysaccharide, apoptosis, mitochondrial membrane potential, intracellular survival

## Abstract

*Brucella suis*, the causative agent of brucellosis, poses a significant public health and animal husbandry threat. However, the role of the alanine racemase (*alr*) gene, which encodes alanine racemase in *Brucella*, remains unclear. Here, we analyzed an *alr* deletion mutant and a complemented strain of *Brucella suis* S2. The knockout strain displayed an unaltered, smooth phenotype in acriflavine agglutination tests but lacked the core polysaccharide portion of lipopolysaccharide (LPS). Genes involved in the LPS synthesis were significantly upregulated in the deletion mutant. The *alr* deletion strain exhibited reduced intracellular viability in the macrophages, increased macrophage-mediated killing, and upregulation of the apoptosis markers. Bcl2, an anti-apoptotic protein, was downregulated, while the pro-apoptotic proteins, Bax, Caspase-9, and Caspase-3, were upregulated in the macrophages infected with the deletion strain. The infected macrophages showed increased mitochondrial membrane permeability, Cytochrome C release, and reactive oxygen species, activating the mitochondrial apoptosis pathway. These findings revealed that alanine racemase was dispensable in *B. suis* S2 but influenced the strain’s rough features and triggered the mitochondrial apoptosis pathway during macrophage invasion. The deletion of the *alr* gene reduced the intracellular survival and virulence. This study enhances our understanding of the molecular mechanism underlying *Brucella*’s survival and virulence and, specifically, how *alr* gene affects host immune evasion by regulating bacterial LPS biosynthesis.

## 1. Introduction

Brucellosis is a zoonotic infection that is characterized by fever and contagious abortion in animals, and it remains one of the most prevalent diseases worldwide. Infection with the *Brucella* pathogen results in chronic debilitating diseases in humans and reproductive disorders in animals, with consequent major public health concerns and severe economic losses in the farming sector [1]. Therefore, a deeper understanding of the pathogenic mechanisms that underlie the *Brucella* infection is of crucial importance for the prevention and control of brucellosis [2].

*Brucella* is a Gram-negative bacterium that grows extracellularly and which replicates within both phagocytic and non-phagocytic host cells [3]. Despite lacking many of the classical virulence factors that other pathogenic bacteria possess, including invasive proteases, exotoxins, capsules, and fimbriae, *Brucella* has evolved a remarkable ability to evade host immunity and cause chronic infection. *Brucella* virulence manifests primarily in the ability to survive and replicate within the phagolysosomal compartment of macrophages [1,4]. The bacterium exhibits smooth and rough colony phenotypes. Smooth strains, such as *B. abortus* 2308, possess intact LPS, inhibit the programmed cell death of infected human and murine macrophages, and, thereby, escape host immunity [5]. In contrast, rough strains lack or produce very low levels of the O-antigen, a long polysaccharide that extends from outer membrane-anchored LPS into the environment. Rough mutants of many bacterial species often exhibit restricted survival within macrophages, which leads to attenuated virulence. Moreover, rough variants of *Brucella* are cytotoxic to macrophages [6].

Alanine racemase (Alr) is a dimeric enzyme that contains pyridoxal 5’-phosphate and catalyzes the interconversion of L-alanine and D-alanine isomers [7]. Alr enzymes comprise biosynthetic and catabolic types: Alr specified by the *alr* gene is involved in cell wall biosynthesis, whereas Alr encoded by *dadX* mediates D-alanine catabolism [8]. Not all bacteria possess both types of *alr* [9]. Certain Gram-negative species, such as *Pseudomonas aeruginosa* PAO1, harbor both the biosynthetic and catabolic versions of the enzyme. In contrast, most Gram-positive bacteria encode only one type of *alr* [10]. For example, *Bacillus pseudofirmus* OF4 and *Streptomyces coelicolor* A3(2) possess only the catabolic and biosynthetic versions of *alr*, respectively [11].

Alr is crucial for bacterial peptidoglycan layer biosynthesis and for cell wall integrity. Therefore, as peptidoglycan biosynthesis does not occur in eukaryotes, the Alr protein is a potential target for the development of novel antibacterial agents that target peptidoglycan production [12,13]. The knockout of the *alr* gene in *Clostridium difficile* resulted in the suppression of spore germination and cellular architecture, accompanied by induced cell wall damage and increased membrane permeability. In *Aeromonas hydrophila*, the knockout of the *alr* gene affected its outer membrane proteins as well as the expression of the genes associated with LPS and adhesins [14,15]. Similarly, the mutation of *alr* loosened the biofilm structure, elevated the extracellular polysaccharide (EPS) content, upregulated the expression of multiple glucosyltransferase genes, and decreased the acid tolerance in *Streptococcus mutans* [16]. These findings suggest that Alr, in addition to catalyzing the isomerization of alanine, plays a key role in bacterial virulence [17]. Despite these intriguing observations, the function of the *alr* gene in *Brucella* has not been investigated fully.

LPS in *Brucella* is a critical virulence factor that is composed of lipid A, core oligosaccharide, and O-antigen [18]. The core oligosaccharide plays a crucial role in connecting lipid A and O-antigen through 2-keto-3-deoxyoctonicacid and quinovosamine [19]. The loci that are involved in LPS production in *Brucella* comprise genes that are required for O-antigen synthesis, core polysaccharide formation, and lipid A synthesis [20] and include the genes for perosamine synthase (*per*), methyltransferase (*wbkC*), mannosyltransferase (*wbkA* and *wbkE*), isomerase/dehydratase (*wbkD*), heptosyltransferase (*wbkF*), and the ABC transport system proteins (*wzm* and *wzt*), among others [21,22,23]. O-antigen is synthesized on the cytoplasmic side of the inner membrane, transported to the periplasmic space via an ABC transport system, and connected to the receptor sugar of the lipid A-core segment [24]. The complete LPS molecule is then transported to the outer membrane and anchored on the outer side of the cell wall.

This study hypothesized that Alr affects the synthesis of LPS anchored on the cell wall surface in *Brucella suis*, the causative agent of brucellosis in pigs. The results showed that the formation of the LPS core oligosaccharide was damaged in the *alr* deletion strain and resulted in an inability to inhibit mitochondrial macrophage apoptosis and a consequent decrease in virulence. The findings supply new insights into the role of the *alr* gene in Gram-negative bacteria and provide important clues for further investigation of the virulence properties of *B. suis*.

## 2. Results

### 2.1. Construction of B. suis Deleted of the alr Gene and a Complemented Strain

To investigate the function of the *alr* gene, we generated a *B. suis* S2 *alr* deletion mutant (Δ*alr*) through homologous recombination using the kanamycin resistance gene as a selection marker to replace the intact *alr* ORF (Figure 1A). PCR amplification yielded a 931 bp upstream homologous arm, a 726 bp downstream homologous arm, and a 1272 bp KanaR fragment targeting the *alr* gene (Figure 1B). To confirm the successful construction of the mutant strain, we designed *alr*-qF/qR primers for qRT-PCR amplification of the *alr* ORF. To ensure the stability of the *alr* mutant strain, we designed upstream primers within the *alr* ORF and downstream primers targeting the KanaR fragment for *alr*-JDTF/JDTR. The mutant strain was passaged for 20 generations, and further confirmation was obtained through PCR analysis (Appendix A), demonstrating the stable inheritance of the deletion strain. The complemented strain (CΔ*alr*) was generated by exogenously expressing the intact *alr* ORF from the Δ*alr* strain using an expression vector with a Flag tag. The generation of the CΔ*alr* strain was validated by qRT-PCR analysis (Figure 1C) and confirmed by Western blotting using mouse anti-Flag antibody (Figure 1D).

### 2.2. Growth Characteristics of B. suis S2 Deleted of the alr Gene

The impact of *alr* gene deletion on the growth and morphology of *B. suis* was assessed by multiple approaches. First, growth was examined in TSA (Figure 1E) which showed no significant differences in the growth rates of wild-type, Δ*alr*, and CΔ*alr* strains. In addition, the growth rate did not change appreciably when D-alanine (0.05–5 mmol/L) was added to the medium (Figure 1F). However, the deletion strain showed significant aggregation in the acriflavine agglutination test after 24 h of static incubation, whereas *B. suis* S2 and CΔ*alr* did not aggregate (Figure 1G). These data suggest that deletion of *alr* changes the membrane composition of *B. suis*.

### 2.3. LPS Characteristics of B. suis S2 Deleted of the alr Gene

In the acidified acridine assay, all three strains exhibited smooth sedimentation without any aggregation, indicating that they were smooth-type *Brucella* strains. The Δ*alr* showed no significant changes, confirming that it still retained the characteristics of a smooth-type *Brucella* strain (Figure 1H). However, further analysis was conducted by extracting the LPS from the three strains and examining their integrity using silver staining. In the silver staining experiment, we observed a significant loss of the core polysaccharide portion in the LPS of Δ*alr* compared with those of *B. suis* S2 and CΔ*alr* (Figure 2A). This indicates that the absence of the *alr* gene affected the integrity of the LPS.

### 2.4. Deletion of alr Affects LPS Synthesis Genes in B. suis S2

The outer cell membrane of *Brucella* contains LPS consisting of lipid A, core oligosaccharide, and O-chain antigen. Lipid A is synthesized on the cytoplasmic side of the inner membrane. Glycosyltransferases attached sugars to form the lipid A-core, which then translocated to the periplasmic side. The O-chain was synthesized on the cytoplasmic side of the inner membrane and connected to the lipid A-core in the periplasmic space. The complete LPS molecule was then transported to the outer membrane.

Finally, the complete LPS molecule was transported to the external side of the outer membrane. We used qRT-PCR to investigate whether deletion of the *alr* gene affected the expression of genes involved in LPS synthesis. The expression of the major genes involved in the synthesis of lipid A, core oligosaccharide, and O-chain was examined at 12 h, 24 h, and 36 h. The expression of *manA*, *manB*, *manC*, *per*, *wbkA*, *wbkC*, *wbkD*, *wbkE*, *wbkF*, *wboA*, *wboB*, *wzm*, and *wzt* was upregulated significantly in the Δ*alr* background compared with the wild-type strain at 12 h, whereas the expression of *wbkB* showed no significant change (Figure 2B). Similarly, the expression of all of these genes was upregulated in the deletion strain after both 24 h (Figure 2C) and 36 h, except that the expression of *per* and *wzm* was unchanged after 36 h (Figure 2D). These results indicate that the expression of the major genes involved in LPS synthesis was upregulated significantly as a consequence of *alr* deletion in *B. suis*.

### 2.5. Deletion of alr in B. suis Affects Cytotoxicity in Macrophages

Lactate dehydrogenase (LDH) is a stable cytoplasmic enzyme that is present in all cell types. The protein is released rapidly into the culture medium when the cell membrane is damaged. We measured LDH levels in the RAW264.7 macrophage cells infected with *B. suis* for 12 h, 24 h, and 48 h to determine whether the deletion of *alr* affected the cell toxicity by the bacterium (Figure 3A). The rates of RAW264.7 cell death differed depending on whether the infection involved the wild-type, ∆*alr*, or C∆*alr* strains and on the length of the infection period. Thus, the death rates were 8.64%, 11.8%, or 8.18% for cells infected with wild-type, ∆*alr*, or C∆*alr* strains, respectively, after 12 h. However, the rates increased in all cases to 13.17%, 17.54%, or 13.3% for wild-type, ∆*alr*, or C∆*alr* strains, respectively, when the infection time was increased to 24 h. The macrophage death rates continued to increase after 48 h of infection to 17.87%, 22.78%, or 17.86% for wild-type, ∆*alr*, or C∆*alr* strains, respectively. In summary, the mortality rates of the ∆*alr*-infected macrophages were significantly increased at all time periods, and this effect was reversed fully by complementation of the deletion in trans (Figure 3A).

### 2.6. The ∆alr Strain Increases the Levels of Apoptosis in RAW264.7 Macrophages

Regulation of host cell apoptosis is a common survival strategy employed by infectious pathogens. For example, smooth *Brucella* species modulate macrophage death by inhibiting the mitochondrial cell death pathway and cysteine protease activation, whereas rough *Brucella* species lack these abilities. The impact of *alr* deletion on cell apoptosis was investigated using flow cytometry with Annexin V and propidium iodide staining of the RAW264.7 cells infected with *B. suis* S2, ∆*alr*, or C∆*alr* (Figure 3D). The early apoptotic rate (24 h) induced by *B. suis* S2 was 2.02%, compared with 6.92% for ∆*alr* and 1.83% for CΔ*alr*. The apoptosis rates after 48 h increased to 13.45%, 16.97%, or 14.98% for the wild-type, ∆*alr*, or C∆*alr* strains, respectively. Thus, apoptosis levels increased over time in each case, but compared with *B. suis* S2 and CΔ*alr,* the Δ*alr* strain enhanced apoptosis of the macrophages at both 24 and 48 h(Figure 3B). These findings suggest that the deletion of *alr* limited the capacity of *B. suis* to inhibit apoptosis of the RAW264.7 macrophages.

### 2.7. The Effects of alr Deletion on Intracellular Proliferation by B. suis S2

In view of the preceding observations concerning the contribution of the *alr* gene to *B. suis* cytotoxicity in the macrophages, we examined the intracellular survival of the wild-type, deletion, and complemented strains. The CFUs of the three strains were determined after infection (200 MOI) of the RAW264.7 macrophages for 6 h, 12 h, 24 h, and 48 h (Figure 3C). The intracellular survival of *B. suis* S2, ∆*alr*, and CΔ*alr* increased with time. However, the survival of ∆*alr* was reduced significantly compared with the wild-type and complemented strains, particularly after 24 h and 48 h (Figure 3C).

*Brucella* is a facultative intracellular parasite, and its virulence manifests mainly in the ability to proliferate within the host cells. The intracellular survival of *Brucella* was evaluated further by staining the nuclei of the RAW264.7 cells with DAPI (blue) and *Brucella* with Alexa Fluor 555 (red) and observing the infected cells by immunofluorescence confocal microscopy. The red fluorescence intensity of the macrophages infected with the ∆*alr* strain was significantly lower compared with the cells infected with *B. suis* S2 or C∆*alr* after both 24 h (Figure 4A) and 48 h (Figure 4B), which indicates a significant reduction in the number of intracellular bacteria in the case of the strain lacking Alr. Thus, the deletion of *alr* reduces the intracellular viability of *B. suis* in the macrophages assessed both by bacterial enumeration and by direct observation of the bacterium within the infected cells.

### 2.8. Mutant Strain ∆alr Depolarization of Mitochondrial Membrane Potential in RAW264.7 Macrophages

The role of Alr in apoptosis of the RAW264.7 cells was examined further by assessing the mitochondrial membrane potential (MMP) of the macrophages after infection with *B. suis* wild-type, deletion, and complemented strains. JC-1 staining was used to evaluate the extent of the mitochondrial damage: red fluorescence represents normal MMP, whereas the intensity of green fluorescence reflects the degree of mitochondrial damage [25]. We observed a significant increase in green fluorescence intensity in the RAW264.7 cells infected with the Δ*alr* strain compared with *B. suis* S2 and C∆*alr* after infection for 24 h. This increase appeared as orange in the merged image (Figure 5A). Thus, the deletion of *alr* induced a significant increase in the MMP damage and led to higher depolarization of the RAW264.7 cells. We also detected MMP after infection for 48 h (Figure 5B), which showed a similar overall trend, with a further increase in the mitochondrial damage over time with the Δ*alr* strain, indicating that the disruption was time-dependent. Mitochondrial damage is a typical feature that is associated with rough strains of *Brucella*, which agrees with the acriflavine agglutination test described above, indicating that *B. suis* Δ*alr* was more prone to clumping than was the wild-type strain.

### 2.9. Mutant Strain ∆alr Induces Accumulation of Reactive Oxygen Species in RAW264.7 Cells

The release of ROS is a characteristic feature that follows mitochondrial damage. We examined the levels of ROS in the RAW264.7 cells infected with *B. suis* S2, CΔ*alr*, or the Δ*alr* strains at 24 h and 48 h post-infection using a DCFH-DA fluorescent probe (Figure 6). This probe is oxidation-sensitive and exhibits no green fluorescence until oxidized by ROS to DCF. Fluorescence microscopy revealed a significant increase in the green fluorescence intensity in the RAW264.7 cells infected with the Δ*alr* strain compared with the wild-type and C∆*alr* strains after both 24 h and 48 h of infection. Furthermore, fluorescence at 48 h was significantly higher than at 24 h in the cells infected with the Δ*alr* strain, which indicates increased ROS production during prolonged infection (Figure 6).

### 2.10. Effect of Deletion of ∆alr on Expression of Apoptotic Proteins in RAW264.7 Cells

Mitochondrial damage typically is associated with the release of Cytochrome C, the inhibition of Bcl2 expression, and the activation of the Bax protein and Caspase family proteins. Therefore, we examined the expression levels of these apoptosis-related proteins to further elucidate the mechanism by which the deletion of *alr* induces apoptosis in the RAW264.7 cells during *B. suis* infection. The RAW264.7 cells were infected with *B. suis* S2, ∆*alr*, or C∆*alr* for 24 h (Figure 7) and 48 h (Figure 8), and whole cell proteins were subjected to Western blotting analysis. Compared with *B. suis* S2 and C∆*alr*, infection with the ∆*alr* significantly increased the release of Cytochrome C, decreased the expression of the upstream Bcl2 protein, inhibited the expression of Bax, and significantly reduced the Bcl2/Bax expression ratio. In addition, the expression of downstream Caspase-9 and Caspase-3 proteins was activated. These results further revealed that *B. suis* which lacked the *alr* gene was impaired in the inhibition of mitochondrial apoptosis in the RAW264.7 macrophage cells.

## 3. Discussion

In this study, we investigated the role of the *alr* gene in *B. suis* S2 using diverse physiological tests combined with gene and protein expression analyses. We found that the deletion of *alr* resulted in impaired LPS synthesis, which diminished the ability of the bacterium to inhibit mitochondrial apoptosis and led to decreased intracellular survival in the macrophages.

The deletion of the *alr* gene did not affect the growth of *B. suis*, even in the absence of the exogenous D-alanine. In contrast, the deletion of two alanine racemase genes and a D-alanine transaminase gene in *Staphylococcus epidermidis* generated a D-alanine auxotrophic strain that required exogenous D-alanine for growth [12]. Variants with mutations in the *alrA* gene in *Mycobacterium tuberculosis* grew on standard minimal media without D-alanine, but the colony morphology of these mutants was drier and more spread out. Alr activity was not detectable in the mutants, which indicated the existence of an alternative D-alanine biosynthesis pathway in *M. tuberculosis. M.tuberculosis*, the causative agent of tuberculosis, possesses a single *alr* gene [26]. Genome-wide transposon insertions and gene knockouts demonstrated that the gene is essential for growth and is absent in humans [13]. Therefore, current tuberculosis treatments involve mainly Alr inhibitors. *Streptococcus pneumoniae* depends on D-alanine as an important precursor for the formation of the peptidoglycan layer, which is a key component of cross-linked polysaccharide chains. As D-alanine is produced from L-alanine by Alr [27,28], inhibiting the formation of the precursor in the peptidoglycan layer in *S. pneumoniae* is a current treatment for middle ear infections by this bacterium. We found that the deletion of *alr* in *B. suis* S2 did not affect normal growth in the absence of D-alanine, which suggests that Alr may not the sole source of D-alanine biosynthesis in *Brucella* and that the protein may not be a plausible drug target to combat brucellosis.

The essentiality of the *alr* gene varies in bacteria, but disrupting the gene has a consistent impact on its bacterial morphology and virulence. Spore germination in *Clostridium difficile* is affected by knocking out *alr* [14]. Knocking out the *alr* gene in *A. hydrophila* resulted in cell wall damage and increased cell membrane permeability [15]. The biomass, cell integrity, EPS synthesis, glucosyltransferase gene expression, acid production, and acid tolerance of *alr* mutant strains of *Streptococcus mutans* were examined. The biofilm formation, cell structure, and EPS synthesis were dose-dependent on D-Alanine [16]. Moreover, the biofilm structure of *alr* mutants was loose, the EPS: bacteria ratio increased, the expression levels of multiple glucosyltransferase gene were upregulated, and the acid tolerance decreased. These and other examples demonstrated that Alr not only catalyzed the conversion of L-alanine to D-alanine but also affected the virulence properties of diverse bacterial species [29].

We found here that the deletion of *alr* affected the synthesis of the core polysaccharide of LPS in *B. suis*. Compared with the parental and complemented strains, the genes involved in the synthesis of O-antigen, core polysaccharide, and lipid A were upregulated in the deletion strain. The *wbk*A and *per* genes of *Brucella abortus* 2308 were involved in the synthesis of the O-antigen of LPS [30,31], and a phosphoglucomutase-deficient derivative of this strain was resistant to the specific Tb phage of smooth *Brucella*, which demonstrated that the *pgm* gene played a crucial role in the glucose synthesis of hexose [32]. Mutations in the *pgm* gene of *Brucella melitensis* caused defects in the core polysaccharide of rough LPS [33,34]. The core oligosaccharide also participated in the resistance to innate immune recognition. Although the structure of the core oligosaccharide was affected by the mutation, the O-antigen did not change. The binding of the *B. melitensis* mutant to the TLR4 accessory receptor MD2 was enhanced, inducing a strong pro-inflammatory response, and the virulence was weakened in the BALB/c mice and DC cells. These observations demonstrated the barrier effect of the core oligosaccharide of *Brucella* LPS against innate immune recognition [35]. Silver staining showed that the core oligosaccharide, which plays a bridging role in connecting lipid A and the O-antigen, was deficient in *B. suis* ∆*alr*. Unexpectedly, the expression of LPS synthesis genes was collectively upregulated in the mutant. Knock-outs of the *alr* gene in *A. hydrophila* showed downregulated expression of the adhesion-related genes; the genes for outer membrane proteins and LPS synthesis were upregulated, and the virulence was weakened in the mice and carp infection models [15]. These observations are in accord with the results presented here. Although seemingly contradictory, we speculate that the core polysaccharide and O-antigen are inhibited when anchored to the cell wall. To compensate for this deficiency, bacteria need to upregulate gene expression to produce more core polysaccharide and O-antigen. Since the absence of *alr* damages the cell wall and outer membrane proteins and LPS exert roles in maintaining cell wall stability, bacteria are forced to produce more components to maintain stability and promote better survival [36].

The survival and proliferation of *Brucella* within host cells are crucial for establishing chronic infection. The mechanisms by which smooth *Brucella* inhibit host cell apoptosis and promote their own proliferation have been studied extensively. Smooth *Brucella* modulate mitochondrial function, for example, by regulating the transition of mitochondrial membranes and the release of Cytochrome C that interacts with the mitochondrial and death receptor apoptotic pathways [37]. In contrast, numerous rough *Brucella* strains are toxic and induce macrophage apoptosis, mainly due to changes in MMP. Here, we found that the ∆*alr* lost the ability to inhibit mitochondrial apoptosis during macrophage infection. Compared with the parental and complemented strains, the deletion derivative increased the level of early apoptosis in the macrophages, which resulted in the depolarization of the MMP and the upregulation of the Cytochrome C protein, which also was accompanied by an increase in ROS. The production of the anti-apoptosis protein Bcl2 decreased, whereas expressions of the pro-apoptosis proteins Bax, Caspase-9, and Caspase-3 were stimulated. These findings suggest that the ∆*alr* mutant has rough *Brucella* intracellular characteristics. Moreover, the deletion of *alr* decreased the intracellular proliferation of *B. suis*, which indicates that the deletion reduced the virulence.

*Brucella* species traditionally have been classified as smooth or rough. However, colonies with both rough and smooth characteristics have been identified (termed RS-type *Brucella*) in certain gene knockout variants [37]. Mutations in the genes involved in the synthesis of core oligosaccharides may also cause the RS phenotype [19]. A mutation in the *wadC* gene disrupted the LPS core polysaccharide, but preserved the LPS polysaccharide and lipid A. These findings indicate the importance of RS-type *Brucella* in vaccine development [38]. In this study, knockout of the *alr* gene damaged the core oligosaccharides, but the yellow pigment test showed that the deletion strain retained smooth characteristics [39]. However, the ∆*alr* strain lost the ability to inhibit mitochondrial apoptosis during infection of the RAW264.7 cells, exhibiting rough *Brucella* characteristics. Therefore, these results suggest that the *alr* deletion strain displays features of RS-type *Brucella*.

In summary, this study demonstrated that the deletion of the gene for *alr* did not impair the ability of *B. suis* to survive in vitro. However, the deletion led to the loss of the core oligosaccharide and caused the upregulation of the LPS synthesis genes, which resulted in the acquisition of RS-type characteristics. Additionally, we found that *B. suis* lacking the *alr* gene was impaired in the inhibition of mitochondrial apoptosis during infection of the macrophage cells. These findings provide new insights into the role of the *alr* gene in Gram-negative bacteria and offer valuable clues for further investigation of the virulence and biological characteristics of *Brucella* sp., as well as developing genetically engineered vaccines against this important pathogen.

## 4. Materials and Methods

### 4.1. Biosafety Statement

Experiments were conducted according to the “Regulations on Biosafety of Pathogenic Microorganism Laboratory” (2004) No. 424 prescribed by the State Council of the People’s Republic of China and were approved by the Biosafety Committee of Northwest A&F University.

### 4.2. Bacterial Strains

*B. suis* S2 (CVCC reference number CVCC70502) was obtained from the Shaanxi Provincial Institute for Veterinary Drug Control (Xi’an, China) and was cultured in tryptic soy broth (TSB, Sigma, St. Louis, MO, USA) with shaking or on tryptic soy agar (TSA, Sigma, St. Louis, MO, USA) at 37 °C. Single bacterial colonies obtained from freshly streaked TSA plates were inoculated into 50 mL TSB and grown at 37 °C for 2 days with gentle shaking. Subsequently, the cultures were diluted 100-fold with TSB and incubated further at 37 °C until exponential phase. The cultures were harvested by centrifugation, resuspended in sterile phosphate-buffered saline (PBS), serially diluted in sterile PBS, and plated onto TSA plates to determine the numbers of viable bacteria. Kanamycin and ampicillin antibiotics were added at 50 and 100 µg/mL, respectively, when required. All procedures involving live *B. suis* S2 strains were conducted in a biosafety level 3 (BSL-3) facility.

### 4.3. Construction of the B. suis ∆alr Deletion and Complemented Strains

A resistance gene replacement procedure was performed to obtain the *B. suis ∆alr* deletion strain. Briefly, the upstream homology arm (931 bp) of the *alr* gene (BSS2_RS14335) was amplified using *alr*-UP-F and *alr*-UP-R primers, and the downstream homology arm (726 bp) was amplified using *alr*-DW-F and *alr*-DW-R primers. The 1272 bp kanamycin resistance (KanR) fragment was amplified from the pEGFP-C1 plasmid using KanR-F and KanR-R primers. The homology arms and KanR fragment were inserted into the pMD19-T vector to generate the recombinant pMD19-T-KanR plasmid, which was transformed into competent *B. suis* S2 cells by electroporation according to standard procedures [40] with selection on kanamycin. Mutant strains were confirmed by PCR using *alr*-qF and *alr*-qR primers. To obtain the complemented strain, we constructed an expression plasmid (pBBARpc) containing a Flag tag [3]. The full-length *alr* open reading frame was amplified using *alr*-F and *alr*-R primers, and the PCR product was cloned into the laboratory’s pre-existing pBBARpc vector to generate the recombinant pBBARpc-*alr* plasmid. *B. suis* ∆*alr* was prepared and transformed with pBBARpc-*alr* using electroporation. Transformants were selected in the presence of ampicillin. Complementation was confirmed by Western blotting using anti-Flag antibody (diluted 1:1000, Absin, Shanghai, China) and by qRT-PCR using *alr*-qF and *alr*-qR primers. Genetic stability of the deletion in ∆*alr* was determined by continuous subculturing for 20 generations followed by amplification using *alr*-JDTF and *alr*-JDTR primers. No reversion was observed, which indicated stable inheritance of ∆*alr*. The sequences of primers used in this study are listed in Table 1. 

### 4.4. Bacterial Aggregation Assay

The bacterial aggregation assay was conducted following a previously published method, with certain adjustments [41]. In summary, *B. suis* S2, Δ*alr*, and CΔ*alr* were cultured until they reached the exponential phase, as previously described. Subsequently, the cultures were placed on test tube racks at room temperature and allowed to stand for 24 h. To determine the extent of bacterial aggregation, the optical density at 600 nm (OD_600_ nm) was measured using a microplate reader (Bio-Rad, Hercules, CA, USA) for 100 µL of the culture before and after the standing period. The percentage of bacterial aggregation was calculated using the following formula:%bacterial aggregation = ([OD_total_ − OD_upperphase_]/OD_total_) × 100

### 4.5. Acriflavine Agglutination Test

The acriflavine agglutination test differentiates smooth and rough bacterial strains. Single colonies of *B. suis* were selected, cultured for 24 h, diluted 1:100, and cultured further for 12 h at 37 °C. Cells were harvested by centrifugation at 8000 rpm for 5 min at 4 °C, the pellet was washed with physiological saline solution three times and then was resuspended in the same buffer. The suspension was diluted to 2 × 10^9^–3 × 10^9^/mL, and 0.5 mL of diluted suspension was mixed with an equal volume of a 0.1% aqueous solution of acriflavine (Aladdin, Shanghai, China) and incubated at 37 °C for 6 h. The mixture was transferred to room temperature for 12 h, and agglutination was observed.

### 4.6. Lipopolysaccharide (LPS) Extraction and Silver Staining

LPS extraction from *Brucella* was performed using the Lipopolysaccharide Extraction Kit (iNtRON Biotechnology, Seongnam-si, Republic of Korea). Briefly, *Brucella* cultures grown for 12 h were centrifuged, and the bacterial pellets were resuspended in 4 mL of Lysis Buffer. After vigorous vortexing for 1 min, 200 μL of chloroform was added and mixed for 10 s. The mixture was incubated at room temperature for 5 min and then centrifuged at 4 °C for 10 min. The resulting supernatant (400 μL) was transferred to a new 1.5 mL centrifuge tube. Next, 800 μL of Purification Buffer was added, followed by incubation at 20 °C for 10 min. After centrifugation at 4 °C for 15 min, LPS particles were washed with 1 mL of 70% ethanol and air-dried. Finally, 30–50 μL of Tris-HCl buffer (pH 8.0) was added to completely dissolve the LPS by vortexing or pipetting. Complete dissolution of the LPS was achieved by incubating in boiling water for 2 min. (All centrifugation steps were performed at 13,000 rpm).

The extracted LPS was subjected to silver staining using the Biyuntian Silver Staining Kit (Biyuntian, Shanghai, China). Briefly, 20 μL of LPS was subjected to SDS-PAGE electrophoresis. The resulting gel was then immersed in approximately 100 mL of fixative and gently shaken at room temperature for 20 min. After discarding the fixative, the gel was treated with 100 mL of 30% ethanol and shaken at room temperature for 30 min. Subsequently, the ethanol was discarded, and 200 mL of double-distilled water was added. The gel was gently shaken at room temperature for 10 min. The water was discarded, and the gel was treated with 100 mL of silver staining sensitizing solution (1×) and gently shaken at room temperature for 2 min. The solution was then discarded, and 200 mL of double-distilled water was added. The gel was gently shaken at room temperature for 1 min. The water was discarded, and the gel was treated with 100 mL of silver staining solution (1×) and gently shaken at room temperature for 10 min. The solution was discarded, and 100 mL of double-distilled water was added. The gel was gently shaken at room temperature for 1 min. Subsequently, the gel was treated with 100 mL of silver staining developing solution and gently shaken at room temperature for 3 min. After discarding the developing solution, the gel was treated with 100 mL of silver staining stop solution (1×) and gently shaken at room temperature for 10 min. The stop solution was discarded, and the gel was treated with 100 mL of double-distilled water, gently shaken at room temperature for 5 min. (The shaking speed for all steps was set at 70 rpm).

### 4.7. Macrophage Cell Infection Assay

RAW264.7 macrophages (obtained from the National Collection of Authenticated Cell Cultures, Shanghai, China) were seeded in 6-well plates at a density of 1 × 10^6^ cells per well or in 24-well plates at a density of 2 × 10^5^ cells per well. The cells were then infected with *B. suis* S2, ∆*alr*, or CΔ*alr* at a multiplicity of infection (MOI) of 200. The infection was allowed to proceed for 4 h at 37 °C in a 5% CO_2_ environment. Subsequently, the macrophages were washed three times with PBS and incubated for an additional 1 h in cell culture medium containing Gentamicin (100 µg/mL) to eliminate any remaining *B. suis* adhered to the macrophages or present in the culture medium. After another round of washing with PBS, the macrophages were further propagated in cell culture medium containing Gentamicin (25 µg/mL) to prevent continuous infection. This time point was considered as 0 h for the subsequent cell infection experiments conducted in this study, following the same protocol. In the cell infection experiments, we included control groups where all steps were performed, except for bacterial infection. The control group followed the same procedures as the experimental groups, except for the introduction of bacterial infection.

### 4.8. Enumeration of B. suis in Infected RAW264.7 Cells

Infected macrophages were washed three times with PBS at 6 h, 12 h, 24 h, and 48 h post-infection and then were lysed using PBS supplemented with 0.5% Triton X-100 for 12 min. The resulting lysates were diluted in PBS, plated on TSA, incubated at 37 °C for 72 h, and colony-forming units (CFU) were counted.

### 4.9. Immunofluorescence Assay

RAW264.7 macrophages were seeded in 24-well plates and infected with *B. suis* strains at MOI 200. The infected cells were washed twice with PBS at 24 h and 48 h post-infection and were fixed with paraformaldehyde (4%) at room temperature for 30 min. Cells were washed three times with PBS and incubated with PBS containing 0.25% Triton X-100 at room temperature for 20 min. Goat anti-*Brucella* polyclonal antibody (diluted 1:1000, A32790, ThermoFisher, Waltham, MA, USA) was used as the primary antibody, followed by donkey anti-goat Alexa Fluor 555 as the secondary antibody (diluted 1:1000, A-21432, ThermoFisher). Nuclei were stained with DAPI for 10 min. The slides were washed four times with PBS after each incubation. Images were captured using an A1R confocal microscope (Nikon, Tokyo, Japan).

### 4.10. Lactate Dehydrogenase Assay

RAW264.7 macrophages were plated in 96-well plates at a density of 1 × 10^4^ cells per well. After 24 h of incubation, the cells were infected with *B. suis* at a multiplicity of infection (MOI) of 200. The post-infection time points analyzed were 12 h, 24 h, and 36 h. The infected cultures were centrifuged, the supernatant was aspirated, and lactate dehydrogenase (LDH) assay detection working solution (60 μL, Beyotime, Shanghai, China) was added to each well; it was mixed and incubated for 30 min at approximately 25 °C in the dark. The absorbance was measured at 490 nm. The cell toxicity rate (%) was calculated as: (absorbance of the treated sample − absorbance of the control sample)/(absorbance of the maximum enzyme activity − absorbance of the control sample) × 100.

### 4.11. Flow Cytometry Analysis

Flow cytometry (Keygen, Nanjing, China) was employed to assess the apoptotic fraction of RAW264.7 cells following *B. suis* infection. Cells were collected at 24 h and 48 h post-infection, centrifuged, and washed three times with PBS. Subsequently, the cells were incubated with 5 µL of FITC-labeled Annexin V and 5 µL of propidium iodide in darkness at room temperature for 15 min. Annexin V stains positive in apoptotic cells, whereas propidium iodide stains positive in necrotic cells. Cells were sorted using a BD FACSAria™ III flow cytometer (BD Biosciences Franklin Lakes, NJ, USA). Data were analyzed using Flowjo 10.0 software with a minimum of 50,000 cells for each determination. 

### 4.12. Measurement of Reactive Oxygen Species Formation 

Intracellular reactive oxygen species (ROS, Beyotime, Shanghai, China) levels were measured by spectrofluorimetry using a 2′,7′-dichlorofluorescein diacetate (DCFH-DA) probe that is oxidation-sensitive and exhibits no fluorescence until oxidized to DCF. RAW264.7 cells (1 × 10^6^ cells/mL) were seeded at 2 mL/well in 6-well plates. Cells were collected 24 h and 48 h after infection and incubated with DCFH-DA (2 μmol/L) for 20 min at 37 °C. Cells were washed three times, and the fluorescence intensity of DCF was determined using with an A1R confocal microscope with excitation and emission wavelengths of 485 nm and 538 nm, respectively. The fluorescence ratio of treated to control values was used to express the relative amount of intracellular ROS production.

### 4.13. Determining the Mitochondrial Membrane Potential Change in RAW264.7 Cells

RAW264.7 cells were collected at the vigorous logarithmic growth phase and seeded into 6-well plates. After 24 h of infection, the medium was aspirated, and the cells were washed with fresh RPMI-1640 (Hyclone, Logan, UT, USA). A working solution of JC-10 (Beyotime, Shanghai, China) stain was added to each well and mixed thoroughly, followed by incubation at 37 °C for 20 min. A 5× JC-10 buffer was diluted to 1× concentration with water and added to each well, which was then placed in an ice bath. The supernatant was aspirated after 20 min, cells were washed twice with buffer, and fresh RPMI-1640 (2 mL) was added. Cells were observed by fluorescence microscopy.

### 4.14. Quantitative Real-Time PCR

*B. suis* S2, ∆*alr*, and C∆*alr* were cultured in TSB until exponential phase and were collected by centrifugation at 4 °C. Total RNA was extracted using TRIzol reagent (Invitrogen, Inc., Carlsbad, CA, USA). The RNA was reverse-transcribed into cDNA using HiScript III RT SuperMix (Vazyme, Nanjing, China) according to the manufacturer’s instructions. Quantitative real-time PCR (qRT-PCR) was performed using ChamQ SYBR qPCR Master Mix (Vazyme) and the Bio-Rad CFX96 Real-Time PCR System (BioRad, Hercules, CA, USA). Relative transcription levels were analyzed using the 2^−ΔΔCt^ method. The results for each target mRNA were normalized to 16S rRNA transcript levels and averaged. Quantitative primers are shown in Appendix A.

### 4.15. Western Blot Analysis

RAW264.7 cells were harvested at 24 h and 48 h post-infection and lysed in lysis buffer on ice for 40 min. The supernatant was obtained by centrifugation at 13,000 rpm for 20 min at 4 °C. Protein concentration was determined using the bicinchoninic acid assay. Total cellular protein was extracted by boiling lysates for 10 min in 5× sodium dodecyl sulfate polyacrylamide gel electrophoresis (SDS-PAGE) loading buffer. Proteins were separated by SDS-PAGE on a 12% polyacrylamide gel and then transferred onto polyvinylidene fluoride membranes. The membranes were blocked for 2 h at room temperature in Tris-buffered saline containing 0.5% Tween-20 (TBST) and 10% skimmed milk and then incubated overnight at 4 °C in blocking solution containing antibodies against Bax (1:2000; Abmart, Shanghai, China), Bcl2 (1:2000; Abmart), cytochrome C (1:5000; Abcam, Cambridge, MA, USA), Caspase-9 (1:1000; Proteintech, Wuhan, China), Caspase-3 (1:1000; Proteintech), or β-actin (1:1000; Proteintech). The membranes were washed four times with TBST for 8 min, incubated for 2 h with the corresponding HRP-conjugated secondary antibody (1:5000; Zhongshan Golden Bridge Biotechnology, Beijing, China), and washed four times in TBST for 8 min. The signal was visualized using an ECL chemiluminescence kit (Beyotime, P0018FS, Shanghai, China) and imaged by the Gel Image System (Tannon Biotech, Shanghai, China).

### 4.16. Statistical Analysis

Statistical analysis was performed with GraphPad Prism software version 6 (GraphPad Software Inc., La Jolla, CA, USA). The significance of the results was determined using one-way or two-way ANOVA. *p*-values < 0.05 were considered statistically significant.

## Figures and Tables

**Figure 1 ijms-24-10744-f001:**
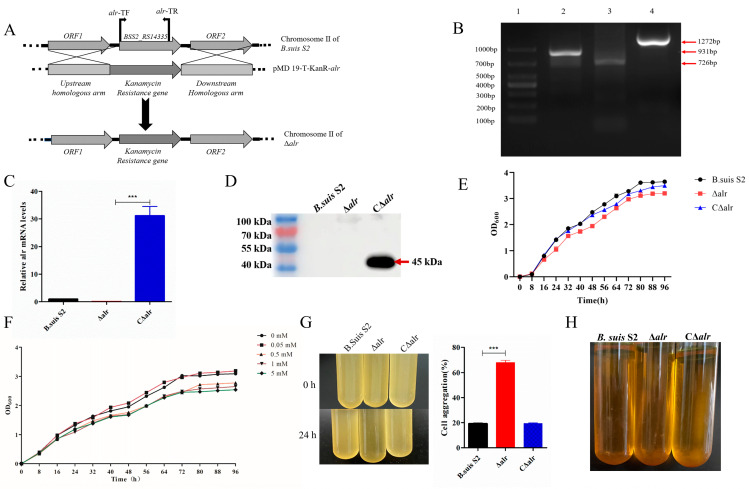
Construction and growth properties of *B. suis* Δ*alr* and complemented strains. (**A**) Schematic of Δ*alr* generation. (**B**) Agarose gel electrophoresis of the Δ*alr* strain. Lanes: 1, upstream homologous arm fragment; 2, downstream homologous arm fragment; 3, KanR fragment. (**C**) qRT-PCR validation of the absence of *alr* gene expression in the deletion strain and restoration of expression in the complemented strain. (**D**) Verification of the CΔ*alr* strain through Western blotting using mouse anti-Flag monoclonal antibody. (**E**) Growth curves of *B. suis* S2, ∆*alr*, and C∆*alr* in TSA medium. (**F**) Growth curves of Δ*alr* with the addition of different concentrations of D-alanine. (**G**) Bacterial aggregation phenotype of the ∆*alr* strain. The three *B. suis* S2 strains were harvested at the exponential phase and left standing at room temperature for 24 h. The OD600 nm of 100 µL of the culture obtained before and after standing was measured to quantify the bacterial aggregation. (**H**) The phenotypes of *B. suis* S2, ∆*alr*, and C∆*alr* were assessed by acridine yellow staining, revealing smooth *Brucella*. All data are presented as the mean ± SEM of three independent experiments. *** *p* < 0.001.

**Figure 2 ijms-24-10744-f002:**
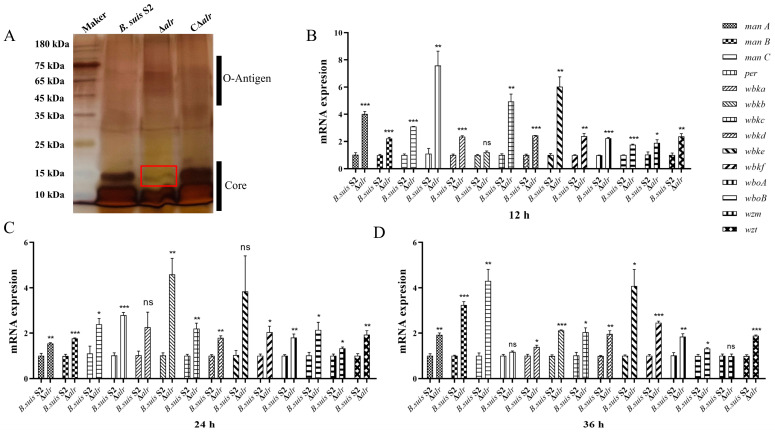
Detection of LPS integrity and expression of LPS-synthesis-related genes. (**A**) LPS stained with silver stain. The red box indicates the portion of the core polysaccharide that is lost in the Δ*alr*. (**B**) Expression levels of *Brucella* genes at 12 h. (**C**) Expression levels of *Brucella* genes at 24 h. (**D**) Expression levels of *Brucella* genes at 36 h. All data are presented as the mean ± SEM of three independent experiments. * *p* < 0.05, ** *p* < 0.01, *** *p* < 0.001, and ns indicates no significant difference.

**Figure 3 ijms-24-10744-f003:**
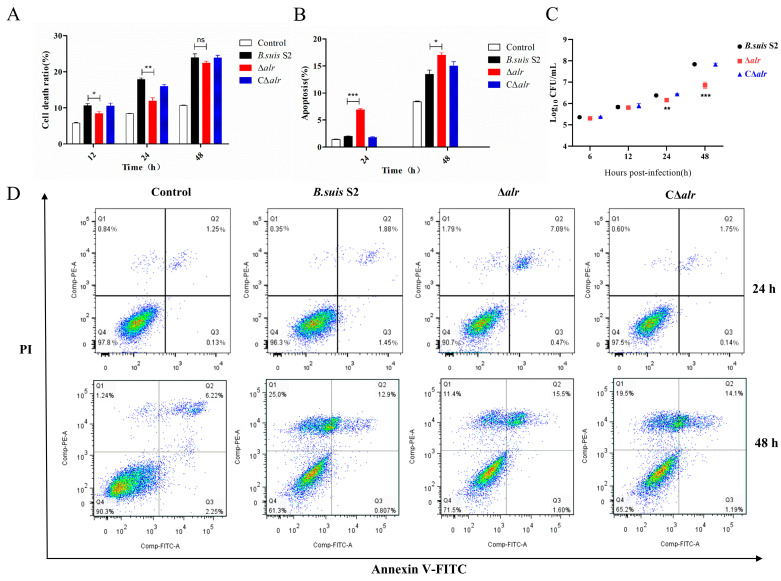
Enhanced intracellular survival of *B. suis* ∆*alr* and inhibition of macrophage apoptosis. (**A**) Intracellular LDH levels were detected at 12 h, 24 h, and 48 h post-infection of RAW264.7 cells with *B. suis* S2, ∆*alr*, and C∆*alr*. (**B**) Statistical analysis of the apoptosis rate of RAW264.7 macrophages. (**C**) CFUs were determined at 6 h, 12 h, 24 h, and 48 h post-infection of RAW264.7 cells with *B. suis* S2, ∆*alr*, and C∆*alr*. (**D**) Flow cytometry analysis of RAW264.7 cells collected at 24 h and 48 h post-infection with *B. suis* S2, ∆*alr*, and C∆*alr*, stained with FITC-labeled Annexin V and propidium iodide. All data are presented as mean ± SEM from three independent experiments. * *p* < 0.05, ** *p* < 0.01, *** *p* < 0.001, and ns, not significant.

**Figure 4 ijms-24-10744-f004:**
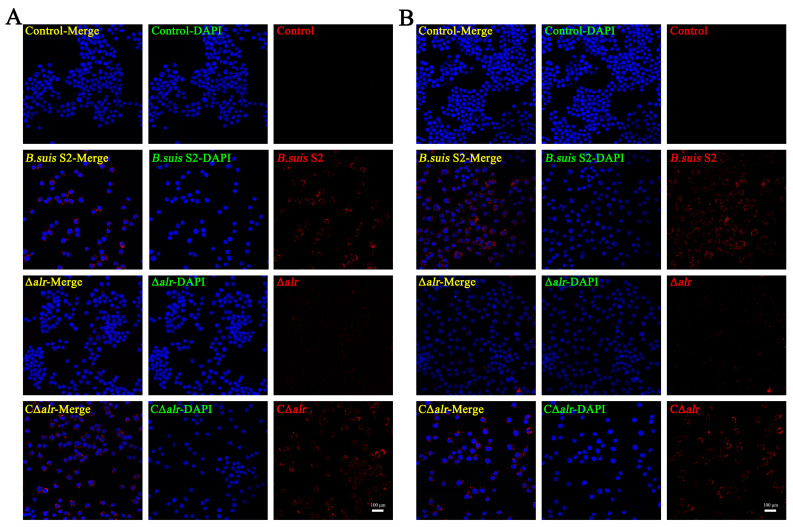
Deletion of *alr* affects the intracellular proliferation of *B. suis* S2. Intracellular proliferation ability of *B. suis* S2 was evaluated in RAW264.7 cells at MOI 200. (**A**) Quantification by immunofluorescence staining of intracellular bacteria after 24 h of infection. (**B**) Quantification by immunofluorescence staining of intracellular bacteria after 48 h of infection. Red fluorescence spots represent Alexa Fluor 555-stained *B. suis* S2, and blue fluorescence represents DAPI-stained cell nuclei. Scale bar, 100 µm. The experiments were conducted in triplicate, and data are presented as means ± SD. Statistical significance was determined by two-tailed Student’s *t*-test for comparison between two groups, and one-way ANOVA followed by post-hoc Bonferroni multiple comparison test for comparison among more than two groups.

**Figure 5 ijms-24-10744-f005:**
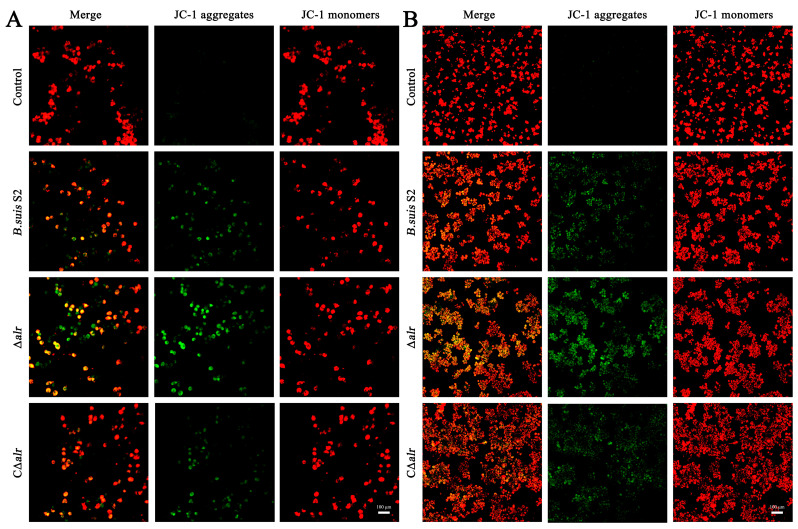
The effect of mutant strain ∆*alr* on MMP in RAW264.7 measured by JC-1 staining intensity. (**A**) RAW264.7 cells were infected with *B. suis* S2, ∆*alr*, and C∆*alr* at MOI 200 for 24 h. (**B**) RAW264.7 cells were infected with *B. suis* S2, ∆*alr*, and C∆*alr* at MOI 200 for 48 h. Scale bar, 100 µm. Fluorescence intensity was observed by confocal microscopy. Red fluorescence represents cells with normal MMP, green fluorescence represents cells with depolarized MMP, and orange indicates the overlap of red and green fluorescence.

**Figure 6 ijms-24-10744-f006:**
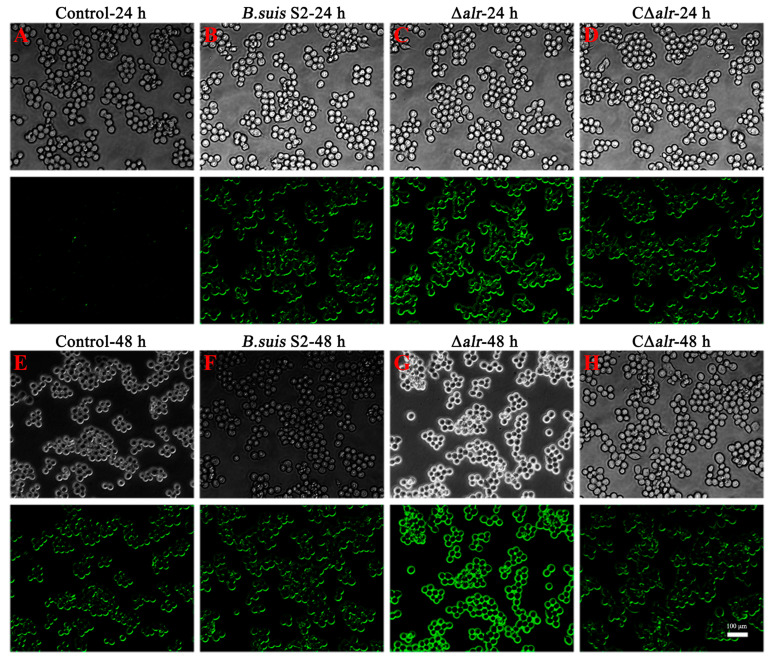
Effect of Δ*alr* on ROS levels in RAW264.7 cells. ROS production in RAW264.7 cells infected with the wild-type strain, ∆*alr* mutant, and complemented strain was detected using DCFH-DA. (**A**) Control group at 24 h. (**B**) *B. suis* S2 group at 24 h. (**C**) Δ*alr* group at 24 h. (**D**) CΔ*alr* group at 24 h. (**E**) Control group at 48 h. (**F**) *B. suis* S2 group at 48 h. (**G**) Δ*alr* group at 48 h. (**H**) CΔ*alr* group at 48 h. Scale bar, 100 µm. The intensity of green fluorescence, which indicates the ROS level, was enhanced with increased ROS levels, as evidenced by an increase in the DCF signal.

**Figure 7 ijms-24-10744-f007:**
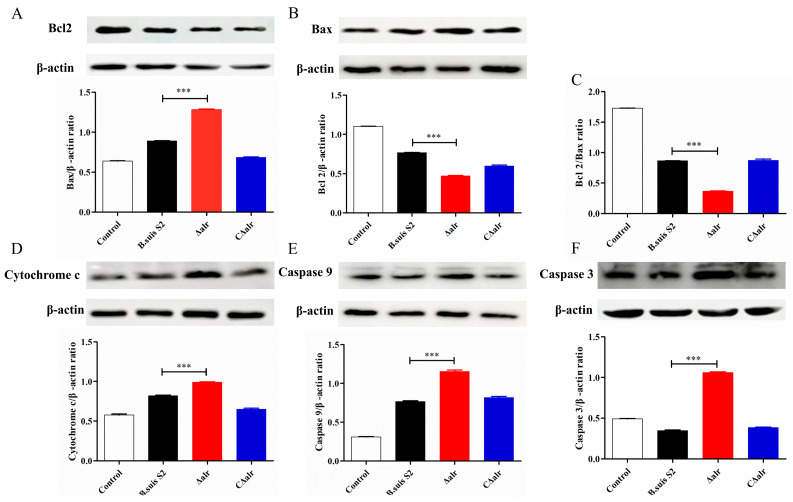
Expression of apoptosis-related proteins in RAW264.7 cells infected with *B. suis* for 24 h detected by Western blotting. (**A**) Ratio of Bcl2 to β-actin protein. (**B**) Ratio of Bax to β-actin protein. (**C**) Ratio of Bcl2 to Bax protein. (**D**) Ratio of Cytochrome C to β-actin protein. (**E**) Ratio of Caspase-9 to β-actin protein. (**F**) Ratio of Caspase-3 to Bax protein. The data are presented as mean ± standard deviation (SD) of triplicate assays. *** *p* < 0.001.

**Figure 8 ijms-24-10744-f008:**
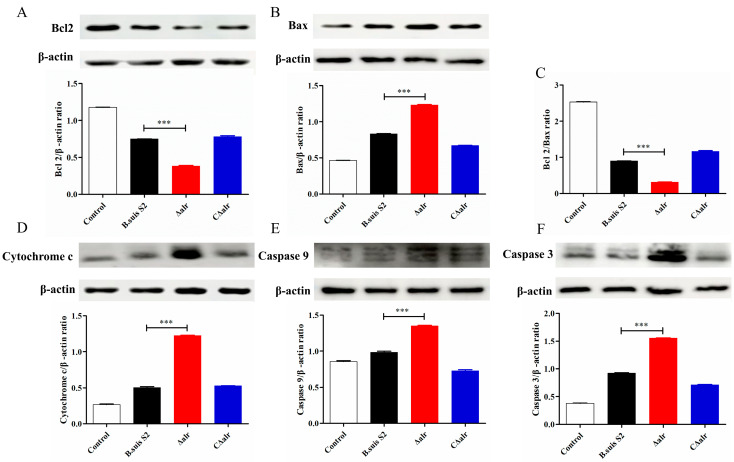
Expression of apoptosis-related proteins in RAW264.7 cells infected with *B. suis* for 48 h detected by Western blotting. (**A**) Ratio of Bcl2 to β-actin protein. (**B**) Ratio of Bax to β-actin protein. (**C**) Ratio of Bcl2 to Bax protein. (**D**) Ratio of Cytochrome C to β-actin protein. (**E**) Ratio of Caspase-9 to β-actin protein. (**F**) Ratio of Caspase-3 to Bax protein. The data are presented as mean ± standard deviation (SD) of triplicate assays. *** *p* < 0.001.

**Table 1 ijms-24-10744-t001:** Primers used in this study.

Primers	Sequence (5′-3′)	Sequence (5’-3′)
*alr*-UP-F/R	TGCGTTACGAGTTCCGCTAATCCTG	CAAGAACTCTGTAGCACCGCACACCCAACGCTTTCCGGGCT
*alr*-DW-F/R	CATTTCCCCGAAAAGTGCCACCTG/ CCACGAGAGGCGTATATGCAG	CTTCACGCCCAGCCCTTCGGCAATC
KanR-F/R	AGCCCGGAAAGCGTTGGGTGT/ GCGGTGCTACAGAGTTCTTG	CTGCATATACGCCTCTCGTGGCAGGTGGCACTTTTCGGGGAAATG
*alr*-JDTF/R	TACGACACCTGGAAGACATC	AAGCCGTTTCTGTAATGAAG
*alr*-qF/R	GCGCTGAAGCCCTTTTTGAA	CGAGACATGCCGGTATCGAA
*alr*-F/R	TTTTATCAGGCTCTGGGAGGGAATAATCTTCACGGTTGAAGTTAA	TGGCACCAGCACAACAGCAGATTACAAGGACGACGATGACAAGTAACGCGGAACCCCTATTTGTTT
*B. suis* 16s-F/R	TATCTAATCCTGTTTGCTCCCC	TGAGTATGGTAGAGGTGAGTGG

## Data Availability

Not applicable.

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
