# Peer review of "Alr Gene in Brucella suis S2: Its Role in Lipopolysaccharide Biosynthesis and Bacterial Virulence in RAW264.7"

_ijms, 2023, doi:10.3390/ijms241310744_

Round 1
Reviewer 1 Report
The research article of Hao et al described the role of Alr in the Brucella. Authors constructed two mutant strains and performed various analyses which elucidate the role of Alr mainly in the biosynthesis of lipopolysaccharides as well as virulence, which is tightly connected with bacterial LPS. I appreciate the laborious experiments which are behind the work. However, I have some minor comments which can improve the presentation of the article:
- Abstract is too long and not all the obtained results must be highlighted in the abstract. Anyway, the abstract for the MDPI is limited to 200 words, so please rewrite it according to the MDPI rules.
- L56 – use a better word instead of In contrast (e.g. Moreover)
- L70-73 – the sentence is long and confusing
- L462 – alr please write italics
- L464 – the dilution of antibody is missing
- L475 – the citation of ‘previously published method’ is missing
- L476 - ‘three strains were cultivated’ – what do authors mean?
- L482 – please correct the formula (0?)
- Please revise all scientific names, they should be written in italics (this mistake occurs very often in the document) – also in the figures (e.g. 1E, 2BCD…) – for both bacterial names and genes names
- Part 4.15 – the chemiluminescence solution is missing
- Figure 1H and 1G – the graph to display the optical density is legible. However, the photos of the cultures are hard to follow. Why are 2 sets of tubes in panel G? For panel H, the optical density will be better representations than the tube photos.
- Part 2.7 – the results can be described more precisely, it is hard to follow the graphs
- Please reconsider the presentation of the figures 2BCD – it is hard to follow and compare the certain gene in the time depending manner. Moreover, please write genes in the legend correctly (italics)
- L179-185 – a brief illustration can improve the explanation and understanding of the composition of Brucella’s LPS (maybe just in the supplement)
- Fig 3 – what does ‘PI’ mean?
Reviewer 2 Report
The manuscript of A. Wang et al. "Alr Gene in Brucella suis S2: Role in Lipopolysaccharide Biosynthesis and Bacterial Virulence in RAW.264.7" is devoted to knockout study of alr gene in B. suis strain S2 and its role to LPS biosynthesis and virulence.
The manuscript is thoroughly written and no problems were found with the text and figures. The only minor issues are missing italization of Latin names and genes in some cases throughout the manuscript (e.g. lines 141, 147, 172, 214-215, 217, 253, 280-281, 299, 300, 315, 321, 462, 494-495, 568).
Also, I have not found supplement figures S1A (line 110) and table 1S (line 603).
Line 543: Shorten Mycobacterium to M.
Line 574: Add FlowJo version.
Line 588: Add manufacturer of RPMI-1640.
The quality of English is pretty good.
